# Deletion of Smooth Muscle O-GlcNAc Transferase Prevents Development of Atherosclerosis in Western Diet-Fed Hyperglycemic ApoE^-/-^ Mice In Vivo

**DOI:** 10.3390/ijms24097899

**Published:** 2023-04-26

**Authors:** Saugat Khanal, Neha Bhavnani, Amy Mathias, Jason Lallo, Shreya Gupta, Vahagn Ohanyan, Jessica M. Ferrell, Priya Raman

**Affiliations:** 1Department of Integrative Medical Sciences, Northeast Ohio Medical University, Rootstown, OH 44272, USA; khanalsaugat7@gmail.com (S.K.);; 2School of Biomedical Sciences, Kent State University, Kent, OH 44242, USA

**Keywords:** O-GlcNAc transferase, O-GlcNAcylation, vascular smooth muscle cells, diabetic atherosclerosis, hyperglycemia

## Abstract

Accumulating evidence highlights protein O-GlcNAcylation as a putative pathogenic contributor of diabetic vascular complications. We previously reported that elevated protein O-GlcNAcylation correlates with increased atherosclerotic lesion formation and VSMC proliferation in response to hyperglycemia. However, the role of O-GlcNAc transferase (OGT), regulator of O-GlcNAc signaling, in the evolution of diabetic atherosclerosis remains elusive. The goal of this study was to determine whether smooth muscle OGT (smOGT) plays a direct role in hyperglycemia-induced atherosclerotic lesion formation and SMC de-differentiation. Using tamoxifen-inducible *Myh11-CreER^T2^* and *Ogt*^fl/fl^ mice, we generated smOGT^WT^ and smOGT^KO^ mice, with and without ApoE-null backgrounds. Following STZ-induced hyperglycemia, smOGT^WT^ and smOGT^KO^ mice were kept on a standard laboratory diet for the study duration. In a parallel study, smOGT^WT^ApoE^-/-^ and smOGT^KO^ApoE^-/-^ were initiated on Western diet at 8-wks-age. Animals harvested at 14–16-wks-age were used for plasma and tissue collection. Loss of smOGT augmented SM contractile marker expression in aortic vessels of STZ-induced hyperglycemic smOGT^KO^ mice. Consistently, smOGT deletion attenuated atherosclerotic lesion lipid burden (Oil red O), plaque area (H&E), leukocyte (CD45) and smooth muscle cell (ACTA2) abundance in Western diet-fed hyperglycemic smOGT^KO^ApoE^-/-^ mice. This was accompanied by increased SM contractile markers and reduced inflammatory and proliferative marker expression. Further, smOGT deletion attenuated YY1 and SRF expression (transcriptional regulators of SM contractile genes) in hyperglycemic smOGT^KO^ApoE^-/-^ and smOGT^KO^ mice. These data uncover an athero-protective outcome of smOGT loss-of-function and suggest a direct regulatory role of OGT-mediated O-GlcNAcylation in VSMC de-differentiation in hyperglycemia.

## 1. Introduction

Cardiovascular disease (CVD) is the leading cause of morbidity and mortality world-wide, claiming about 19 million lives in 2020 [1]. CVD has continued to pose tremendous economic burden on national health care expenditure, with an estimated total cost of $378 billion in 2017–2018. Atherosclerosis, one of the major players in cardiovascular anomalies, results in occlusion of the affected arteries triggering multiple complications including ischemic stroke, myocardial infarction, heart failure and angina [2]. Despite the well-accepted contribution of lipids to atherosclerosis, current lipid-lowering therapies including statins and PCSK9 inhibitors may offer limited benefit against reduction of major adverse cardiovascular events. Moreover, the emergence of statin intolerance and associated risks of hyperglycemia coupled with the escalated costs of PCSK9 inhibitors have raised added concerns about the risk–benefit ratio of these agents [3,4].

Vascular disease is the most prevalent and deadliest of all health concerns affecting diabetic patients in our society. Epidemiological studies indicate that diabetic patients and individuals with impaired glucose tolerance have a 2-4-fold increased risk for development of atherosclerosis compared to individuals without diabetes [5]. Multiple clinical studies and trials including animal data have shown that elevated plasma glucose levels, a hallmark of diabetes, have profound proatherogenic properties independent of hyperlipidemia [6,7,8]. Taken together, these findings highlight hyperglycemia as an independent risk factor for initiation and progression of macrovascular complications. Yet optimal glycemic control alone does not effectively lower the risks of atherosclerosis in long-standing diabetic patients [9]. Indeed, previous studies have provided evidence for the role of chronic hyperglycemia and duration of diabetes in the development of atherosclerosis [10,11]. This has led to the idea that deranged glucose metabolism following chronic hyperglycemia may be responsible for accelerated atherosclerotic complications associated with diabetes. However, the inherent molecular basis for diabetes-induced vasculopathy remains incompletely understood.

In healthy individuals, vascular smooth muscle cells (VSMC), residing in the medial layer of the vascular wall, play a key role in maintaining the normal vascular tone, enabling effective flow of arterial blood throughout the body. In diabetic patients, chronic exposure to elevated circulating glucose concentrations makes these individuals highly prone to increased VSMC migration and proliferation [12]. De-differentiation of SMC into the synthetic phenotype with augmented migratory and proliferative properties, accompanied with reduced contractile gene expression, is a key event central in the evolution of atherosclerosis [13,14,15]. Despite extensive studies, a complete molecular understanding of how hyperglycemia regulates this process is currently obscure.

O-GlcNAcylation is an important post-translational modification that involves the attachment of an O-linked N-acetylglucosamine (O-GlcNAc) moiety to serine and threonine residues of numerous cytoplasmic, nuclear and mitochondrial proteins [16,17]. This process is tightly regulated by two key enzymes, O-linked N-acetylglucosaminyltransferase (OGT) and O-linked N-acetyl-glucosaminidase (OGA). A plethora of data indicates that O-GlcNAc protein modification is a key regulatory mechanism of intracellular glucose signaling [18]. It is widely accepted that, under conditions of increased nutrient availability such as diabetes, cellular O-GlcNAc levels are profoundly elevated [19,20]. Indeed, several studies, including our earlier work, have shown that hyperglycemia positively correlates with increased O-GlcNAcylation in various cells and tissues [18,21,22]. A growing literature further reinforces the notion that O-GlcNAcylation is a putative pathogenic contributor to diabetes and related vascular complications [23,24,25,26,27]. Previous studies have shown that incubation of vascular cells including VSMC and endothelial cells with high glucose in vitro, reflective of the diabetic milieu, activates the hexosamine biosynthetic pathway (HBP), resulting in sustained increase in protein O-GlcNAcylation. Augmented O-GlcNAc protein modification, in turn, mediates the upregulation of numerous genes associated with atherosclerosis, including TSP-1, TGF-β, PAI-1 and NF-kB [28,29,30,31]. Earlier studies have also shown that diabetic patients may develop atherosclerotic plaques with elevated O-GlcNAc levels [32]; incidence of calcified plaques is also profoundly enhanced in those individuals [33,34].

We previously demonstrated that protein O-GlcNAcylation directly associates with increased proliferation of human aortic smooth muscle cells (HASMCs) in response to high glucose in vitro [22]. In a subsequent study, we further reported that protein O-GlcNAcylation and augmented OGT expression, a key controller of O-GlcNAc signaling, correlate with increased atherosclerotic lesion formation and proliferative SMC lesion abundance in hyperglycemic ApoE^-/-^ mice in vivo [35]. Recent studies have further shown that OGT, the sole enzyme mediating addition of O-GlcNAc residues to proteins, plays a crucial role in regulation of diabetic vascular calcification and diabetic wound recovery [26,36]. However, the fundamental contribution of OGT as a driver of atherosclerotic lesion formation in diabetes has not been previously explored.

Therefore, the primary objective of the current study was to determine whether smooth muscle OGT plays a direct role in the development of hyperglycemia-induced atherosclerosis and SMC de-differentiation in diabetes. Our data demonstrate an athero-protective consequence of SMC-specific OGT deletion in Western diet-fed hyperglycemic ApoE^-/-^ mice in vivo.

## 2. Results

### 2.1. Validation of Inducible SMC-Specific OGT Knockout Mice

To study the role of smooth muscle OGT, we generated inducible SMC-specific OGT knockout mice (Figure 1A) by crossing female homozygous *Ogt* floxed mice with Tmx-inducible *Myh11-CreER^T2^* (hemizygous Cre^tg^) male mice, expressing CreER^T2^ under control of the mouse smooth muscle myosin heavy chain 11 promoter. Figure 1B shows genotyping data confirming generation of OGT^fl/Y^/Cre^tg^ and OGT^+/Y^/Cre^tg^ mice. Mice were injected with Tmx or corn oil, as described in Methods for Cre recombinase activation. This was followed by collection of the aortae and left ventricular tissue at least 3 weeks post-Tmx. Immunoblotting confirmed a significant reduction in OGT expression that was noted only in aortic lysates prepared from Tmx-treated OGT^fl/Y^/Cre^tg^ mice. Specifically, OGT expression was reduced by about 70% in aortic vessels derived from Tmx-treated OGT^fl/Y^/Cre^tg^ mice vs. Tmx-treated OGT^+/Y^/Cre^tg^ littermates (Figure 1C, left panel; *p* < 0.0001). In contrast, OGT expression was unaffected in the left ventricle (LV) tissue lysates prepared from these mice genotypes under similar experimental conditions. These results validated successful generation of inducible SMC-specific OGT knockout murine model (smOGT^KO^).

### 2.2. Metabolic Phenotype of SMC-Specific OGT Knockout Mice under Basal and STZ-Induced Hyperglycemia

To induce hyperglycemia reflective of diabetes, smOGT^WT^ and age-matched smOGT^KO^ mice were treated with STZ or sodium citrate (vehicle control) intraperitoneally, as described in Methods. Body weight and random blood glucose levels were monitored every two weeks in all animals until harvest at 16-wks-age (Figure 1D). As shown in Figure 2A, no significant differences in body weights were observed in smOGT^KO^ vs. smOGT^WT^ mice treated with or without STZ. Random blood glucose monitoring revealed significantly elevated glucose levels (>250 mg/dL) in STZ-treated mice genotypes, as early as 10 days following the final dose of STZ administration. Intraperitoneal glucose tolerance test (IPGTT) was conducted to examine glucose responsiveness in the mice genotypes. As expected, STZ administration significantly lowered the ability of the mice to handle an intraperitoneal glucose load and this effect was observed in both smOGT^WT^ and smOGT^KO^ mice (Figure 2B). Moreover, no statistically significant difference in plasma total cholesterol and total triglyceride levels were noted between smOGT^WT^ and smOGT^KO^ mice under STZ- and non-STZ treated conditions (Figure 2C,D). In addition, echocardiographic measurements revealed unaltered Ejection Fraction (%EF), Fractional Shortening (%FS), left ventricular internal diameter (LVID) and left ventricular volume (LV vol) in smOGT^WT^ vs. smOGT^KO^ mice, and this effect was consistent across both non-hyperglycemic and STZ-induced hyperglycemic mice (Figure 2E–H). Indirect calorimetry using CLAMS demonstrated no significant difference in oxygen consumption (VO_2_, Figure 3A), carbon dioxide production (VCO_2_, Figure 3B), RER, a measure of fuel usage (Figure 3C) and other metabolic parameters including physical activity (Figure 3D), energy expenditure (Figure 3E) and food intake (Figure 3F) between non-hyperglycemic and hyperglycemic mice with or without SMC-specific OGT deletion. Furthermore, assessment of fat and lean mass using Echo MRI did not show any difference in adiposity between any of the mice treatment groups (Figure 3G). Together, these results clearly demonstrate that SMC-specific loss of OGT does not adversely affect the metabolic phenotype and cardiac function of STZ-induced hyperglycemic mice in vivo.

### 2.3. Reduced O-GlcNAc Protein Expression Increases SM Contractile Marker Expression in the Aortic Vasculature of STZ-Induced Hyperglycemic Mice with SMC-Specific OGT Deficiency

To determine whether smooth muscle OGT plays a direct role in VSMC differentiation in diabetes, aortic lysates derived from non-hyperglycemic and STZ-induced hyperglycemic smOGT^WT^ and smOGT^KO^ mice were subjected to immunoblotting to measure ACTA2 and LMOD1 expression, markers of smooth muscle contractile phenotype. As shown in Figure 4A,B, OGT and O-GlcNAc protein expression were upregulated in the aortic vessels of smOGT^WT^ mice following STZ-induced hyperglycemia. In contrast, SMC-specific OGT deletion significantly reduced OGT and O-GlcNAc protein levels in the aortic vasculature of smOGT^KO^ vs. smOGT^WT^ mice under non-hyperglycemic and hyperglycemic conditions. Specifically, while STZ-induced hyperglycemia increased OGT and O-GlcNAc protein expression by 2-fold in smOGT^WT^ mice (*p* < 0.0001 vs. non-hyperglycemic smOGT^WT^), SMC-specific lack of OGT decreased OGT expression and protein O-GlcNAcylation in smOGT^KO^ mice (*p* < 0.0001 compared to smOGT^WT^), and this effect was observed both basally and in response to hyperglycemia. Interestingly, in the aortic vasculature of STZ-treated smOGT^KO^ mice, hyperglycemia failed to further increase the residual OGT and O-GlcNAc protein expression. This was likely due to be to the SMC-restricted genetic ablation of OGT blocking O-GlcNAc signaling in smooth muscle cells, a major cell type in the vessel wall. While OGA expression showed an increasing trend in aortic lysates from hyperglycemic vs. non-hyperglycemic smOGT^WT^ mice, this difference did not reach statistical significance (*p* = 0.054). Moreover, there was a significant decrease in OGA expression in STZ-treated smOGT^KO^ mice (Figure 4C, *p* < 0.05 vs. STZ-treated wild-type mice with intact OGT). Importantly, SMC-specific OGT loss and reduced protein O-GlcNAcylation increased SM contractile marker expression, reflective of SMC differentiation. Briefly, in aortic vessels of hyperglycemic smOGT^KO^ mice, there was >2-fold increase in ACTA2 expression compared to hyperglycemic smOGT^WT^ mice (*p* < 0.0001, Figure 4D). Likewise, SMC-specific lack of OGT augmented LMOD1 expression in both non-hyperglycemic and hyperglycemic smOGT^KO^ mice vs. animals with intact OGT (*p* < 0.005, Figure 4E). Further, real-time PCR demonstrated a statistically significant increase in *Myh11* mRNA expression in smOGT^KO^ vs. smOGT^WT^ mice in response to STZ-induced hyperglycemia (*p* < 0.05; Figure 4F). Collectively, these data clearly demonstrate a direct regulatory role of smooth muscle OGT on VSMC de-differentiation in response to hyperglycemia.

### 2.4. Effect of SMC-Specific OGT Deletion on Glycemic Index and Plasma Lipid Levels in Western Diet-Fed ApoE^-/-^ Mice In Vivo

To interrogate whether smooth muscle OGT plays a direct role in development of atherosclerosis, we next generated smOGT^KO^ mice on an ApoE^-/-^ atherosclerotic background. Immunoblotting of aortic lysates revealed a significant decrease in OGT and O-GlcNAc protein expression in Tmx-treated Cre^tg^/OGT^fl/Y^/ApoE^-/-^ mice compared with age-matched Tmx-injected Cre^tg^/OGT^+/Y^/ApoE^-/-^ littermate mice (*p* < 0.0001, Appendix A), validating the smOGT^KO^/ApoE^-/-^ murine model. Mice were subjected to a Western diet feeding regimen, as illustrated in Figure 5A. Western diet feeding significantly increased fasting blood glucose levels (~2-fold; *p* < 0.0001) in smOGT^WT^ApoE^-/-^ mice vs. standard chow-fed smOGT^WT^ApoE^-/-^ mice (Appendix A), reflective of hyperglycemia characteristic of diabetes. Notably, hyperglycemia was accompanied with a significant elevation in O-GlcNAc protein expression in aortic vessels isolated from the Western diet-fed smOGT^WT^ApoE^-/-^ vs. standard chow-fed smOGT^WT^ApoE^-/-^ mice (Appendix A). Interestingly, under these experimental conditions, SMC-specific OGT ablation had no effect on the body weight, fasting blood glucose, glucose tolerance and plasma lipid levels in Western diet-fed hyperglycemic smOGT^KO^ApoE^-/-^ compared with hyperglycemic smOGT^WT^ApoE^-/-^ mice with intact OGT (Figure 5B–G). These results demonstrate that SMC-specific lack of OGT does not affect the metabolic parameters of Western diet-fed ApoE^-/-^ in vivo.

### 2.5. SMC-Specific OGT Deficiency Impedes Lesion Burden in Western Diet-Fed Hyperglycemic ApoE^-/-^ Mice

Using aortic root morphometry, we analyzed atherosclerotic lesion formation in hyperglycemic smOGT^WT^ApoE^-/-^ and smOGT^KO^ApoE^-/-^ mice following 6–7 weeks of Western diet feeding. As shown in Figure 6A, Oil red O (ORO) staining of aortic root sections revealed large lipid-filled lesions in smOGT^WT^ApoE^-/-^ mice. In contrast, SMC-specific OGT deletion significantly attenuated the lipid burden in smOGT^KO^ApoE^-/-^ aortic roots (Figure 6A). Quantification of the ORO-positive area demonstrated a statistically significant reduction (*p* < 0.0001) in lipid-filled lesions formed in the aortic sinus of smOGT^KO^ApoE^-/-^ mice vs. smOGT^WT^ApoE^-/-^ with intact OGT, in response to Western diet-induced hyperglycemia (Figure 6C). To further assess the plaque area in the mice, we next performed H & E staining of the aortic root sections. As shown in Figure 6B, hyperglycemic smOGT^WT^ApoE^-/-^ mice with intact OGT showed a remarkable plaque area that was consistent with the increased lipid burden in these animals, as compared with hyperglycemic ApoE^-/-^ mice lacking SMC-specific OGT. Specifically, quantification of H & E staining revealed >3-fold decrease in aortic root lesion area in smOGT^KO^ApoE^-/-^ vs. smOGT^WT^ApoE^-/-^ with intact OGT following Western diet feeding (*p* < 0.0001; Figure 6D). Taken together, these data clearly demonstrate an athero-protective effect of SMC-specific OGT deficiency in Western diet-fed hyperglycemic ApoE^-/-^ mice in vivo.

### 2.6. SMC-Specific OGT Deficiency Abrogates Inflammatory and Smooth Muscle Cell Lesion Abundance in Western Diet-Fed Hyperglycemic ApoE^-/-^ Mice

Aortic root lesions formed in Western diet-fed hyperglycemic smOGT^WT^ApoE^-/-^ mice (with intact OGT) revealed remarkable leukocyte infiltration and SMC content, as shown via immunohistochemistry. In contrast, SMC-specific OGT ablation reduced both leukocyte and SMC abundance within aortic root lesions of Western diet-fed smOGT^KO^ApoE^-/-^ mice (Figure 7A,B). Specifically, CD45 expression depicting leukocyte content was significantly diminished in the aortic root lesions of smOGT^KO^ApoE^-/-^ mice lacking smooth muscle OGT, reflecting a reduced inflammatory lesion burden in these animals (3-fold vs. smOGT^WT^ApoE^-/-^, *p* < 0.005; Figure 7C). Similarly, immunofluorescence quantification revealed attenuated ACTA2-positive area in aortic root sections derived from Western diet-fed smOGT^KO^ApoE^-/-^ genotypes, illustrating attenuated ACTA2 abundance in these animals (*p* < 0.0001 vs. smOGT^WT^ApoE^-/-^; Figure 7D). In each case, comparable tissue sections incubated with an isotype-specific IgG antibody (instead of the corresponding species-specific primary antibody) displayed negligible background staining, confirming staining specificity for each antibody. Together, these results clearly demonstrate that SMC-specific lack of OGT blocks inflammatory and SMC lesion invasion in hyperglycemic ApoE^-/-^ mice in vivo.

### 2.7. Lack of SMC-Specific OGT Increases SM Contractile Marker Expression While Reducing Inflammatory and Proliferative Marker Expression in Western Diet-Fed ApoE^-/-^ Aortic Vasculature

To delineate the role of smooth muscle OGT in VSMC differentiation and inflammatory response, aortic lysates from Western diet-fed smOGT^KO^ApoE^-/-^ and smOGT^WT^ApoE^-/-^ mice were subjected to immunoblotting. As shown in Figure 8A,B,D,E, SMC-targeted OGT deletion significantly augmented LMOD1 (>2-fold; *p* < 0.005) and ACTA2 (~3-fold; *p* < 0.05) expression (SM contractile markers) in aortic vessels of Western diet-fed smOGT^KO^ApoE^-/-^ mice compared to smOGT^WT^/ApoE^-/-^ with intact OGT. Under similar experimental conditions, loss of SMC-specific OGT reduced the expression of PCNA (proliferation marker) accompanied with diminished ERK phosphorylation in smOGT^KO^ApoE^-/-^ mice (Figure 8C). Specifically, there was >3-fold decrease in PCNA expression coupled with ~70% attenuation in pERK/tERK expression noted in the aortic vasculature of smOGT^KO^ApoE^-/-^ compared with smOGT^WT^ApoE^-/-^ mice following Western diet feeding (*p* < 0.05, Figure 8F,G). Real-time PCR experiments further revealed that SMC-specific OGT deficiency significantly augmented the relative mRNA expression of classical SM contractile genes in Western diet-fed smOGT^KO^ApoE^-/-^ aortic vessels compared to age-matched smOGT^WT^ApoE^-/-^ mice (Figure 8H). Specifically, both smooth muscle actin (*Acta2*) and calponin (*Cnn1*) mRNA levels were elevated in ApoE^-/-^ mice lacking SMC-specific OGT (*p* < 0.05 vs. ApoE^-/-^ with intact OGT). Moreover, under these experimental conditions, increased SM contractile gene expression was accompanied with downregulation of *Il6* and *Il1β* mRNA levels in mice lacking smooth muscle OGT. Specifically, both *Il6* and *Il1β* mRNA expression were attenuated >45% in aortic vasculature of Western diet-fed smOGT^KO^ApoE^-/-^ vs. ApoE^-/-^ mice with intact OGT (Figure 8H). Together, these data clearly demonstrate that SMC-specific OGT deletion promotes SMC differentiation in aortic vessels of hyperglycemic ApoE^-/-^ mice, evidenced via upregulated SM contractile and reduced inflammatory and proliferative marker expression.

### 2.8. SMC-Specific OGT Deletion Attenuates YY1 and SRF Expression in Aortic Vessels of Hyperglycemic Mice In Vivo

Concomitant to increased SM contractile marker expression, immunoblotting of aortic lysates prepared from smOGT^KO^ApoE^-/-^ mice revealed attenuated YY1 and SRF expression (transcriptional regulators of SM contractile genes). Specifically, YY1 and SRF protein expression were significantly diminished in the vascular walls of Western diet-fed ApoE^-/-^ mice with SMC-specific OGT deletion (*p* < 0.0001 vs. smOGT^WT^ApoE^-/-^; Figure 9A,B). Consistently, both SRF and YY1 protein expression were markedly reduced in the aortic vessels of STZ-induced hyperglycemic smOGT^KO^ mice. Specifically, hyperglycemia upregulated YY1 expression in smOGT^WT^ aortic vessels (2.9-fold vs. normoglycemic smOGT^WT^; *p* < 0.0005). In contrast, lack of SMC-specific OGT profoundly diminished YY1 expression in hyperglycemic smOGT^KO^ mice (3.6-fold vs. hyperglycemic smOGT^WT^; *p* < 0.0001, Figure 9D). Similar to these results, SMC-specific OGT deficiency downregulated SRF expression in smOGT^KO^ mice compared with smOGT^WT^ with intact OGT, and this effect was observed under both basal and hyperglycemic conditions (3–4-fold, *p* < 0.0005, Figure 9C). Interestingly, while hyperglycemia increased YY1 expression in the vascular walls of smOGT^WT^ mice, SRF expression was elevated in smOGT^WT^ aortic lysates both basally and in response to STZ-induced hyperglycemia, with no significant difference between these treatment groups. Finally, as shown via real-time PCR, *Myocd* mRNA expression was upregulated in aortic tissues of STZ-induced hyperglycemic smOGT^KO^ mice lacking OGT compared with hyperglycemic smOGT^WT^ genotypes with intact OGT (*p* < 0.05; Figure 9E). Collectively, these results demonstrate that smooth muscle OGT modulates YY1 and SRF expression under hyperglycemic conditions.

## 3. Discussion

The present study provides novel evidence for a causal role of smooth muscle OGT and O-GlcNAcylation in hyperglycemia-induced atherosclerosis. Although earlier work has reported that OGT-mediated O-GlcNAcylation promotes diabetic vascular calcification [26], the conceptual link between SMC-derived OGT and hyperglycemia-driven lesion pathogenesis has not been previously explored. In this report, we provide the first demonstration that smooth muscle OGT-dependent mechanism(s) drive atherosclerotic lesion formation stimulated by hyperglycemia. Notably, our data suggests that OGT-mediated O-GlcNAcylation regulates VSMC de-differentiation in response to hyperglycemia.

In the current work, loss of SMC-specific OGT did not affect either body weight, glycemic index, or lipid levels in hyperglycemic mice in vivo. Metabolic parameters, including oxygen consumption (VO_2_), carbon dioxide production (VCO_2_), physical activity, food intake, and energy expenditure, were also unaffected in mice lacking smooth muscle OGT, and this effect was observed under both hyperglycemic and non-hyperglycemic conditions. Our results concur with an earlier report showing comparable body weight, oxygen consumption, food intake, and physical activity in liver-specific OGT knockout mice compared to wild type animals with intact OGT signaling [37].

It is well established that protein O-GlcNAcylation can affect numerous physiological and pathophysiological functions via regulation of transcription, metabolism, cellular localization, mitochondrial function, protein turnover and protein stability, as well as autophagy [38]. Previous studies have shown that increased O-GlcNAcylation of cardiac proteins may contribute to the adverse effects of diabetes in the heart and cardiovascular system; paradoxically, transient increase in cardiac protein O-GlcNAcylation may have beneficial effects on the heart [39]. Specifically, OGT ablation in cardiomyocytes was reported to cause dilated cardiomyopathy with systolic and diastolic dysfunction [27,40]. Conversely, recent studies using in vitro and in vivo murine models of ischemia-reperfusion injury have demonstrated cardioprotective responses of elevated protein O-GlcNAcylation [41,42]. In an inducible cardiomyocyte-specific OGT knockout murine model, it was further shown that conditional OGT ablation in cardiomyocytes leads to progressive ventricular dysfunction in the adult mice, supporting a critical role of cardiomyocyte OGT in mammalian heart maturation [43]. In the current study, inducible SMC-specific OGT deletion did not initiate an adverse cardiac phenotype in our animals. This data validates the smooth muscle specificity of the Cre murine driver *Myh11-CreER^T2^* employed, devoid of any off-target effects.

Clinical and animal data, including genetic association studies, have linked hyperglycemia, a hallmark feature of Type 1 and Type 2 diabetes, with increased O-GlcNAc signaling [20,44,45]. Extensive literature, including our earlier work, indicates that augmented protein O-GlcNAcylation is a putative trigger for diabetes-related complications [22,23,24]. In the present study, we have demonstrated that SMC-specific loss of OGT resulting in reduced O-GlcNAcylation prevents atherosclerotic lesion formation in ApoE^-/-^ mice induced by Western diet feeding. It is important to note that Western diet feeding increased fasting blood glucose levels in the ApoE^-/-^ mice, prompting a diabetic phenotype in these animals. Such an increase in glycemic levels was accompanied by augmented protein O-GlcNAcylation in the vasculature (Appendix A). These data are in accordance with earlier reports [46,47] revealing elevated fasting glucose and insulin levels in ApoE^-/-^ mice following a Western diet feeding regimen. Importantly, we show that SMC-specific OGT loss impedes lipid burden and plaque area in hyperglycemic ApoE^-/-^ mice, in the absence of significant changes in blood glucose and plasma lipid levels, compared with hyperglycemic ApoE^-/-^ with intact OGT. These results postulate a direct regulatory role of OGT-mediated O-GlcNAcylation in atherosclerotic lesion pathogenesis, independent of glucose or lipid alterations. It is also noteworthy that normal chow feeding did not elevate blood glucose levels to the hyperglycemic range in smOGT^WT^ApoE^-/-^ mice. Likewise, age-matched smOGT^KO^ApoE^-/-^ mice did not develop hyperglycemia in response to normal chow feeding. Under such conditions of normoglycemia, no significant difference was observed in lesion area, lipid burden and SM contractile marker expression (Appendix A) between normal chow-fed smOGT^KO^ApoE^-/-^ vs. smOGT^WT^ApoE^-/-^ mice (with intact OGT). Together, these data further support the concept that the athero-protective phenotype of SMC-specific OGT deletion is specific to only high glucose conditions, relevant to diabetes.

Several lineage tracing studies have established the contribution of vascular smooth muscle cells to atherosclerotic lesion formation [48,49,50]. De-differentiation of vascular smooth muscle cells is characterized by enhanced migratory, secretory and proliferative properties concomitant to loss of the contractile gene expression. Diabetic patients have an increased propensity for VSMC migration and proliferation, features characteristic of VSMC de-differentiation from a ‘quiescent’ contractile state to a ‘synthetic’ proliferative phenotype. In a recent study, a positive correlation was noted between SMC de-differentiation and OGT-mediated O-GlcNAcylation in PDGF- and rapamycin-treated coronary SMC [51]. Further, in a murine model of carotid artery ligation, it was shown that, while vascular injury increased OGT and O-GlcNAc protein expression, SMC-specific OGT loss-of-function reduced neointima formation [52]. Consistent with these earlier reports, in the current work we have shown that SMC-specific lack of OGT resulting in reduced protein O-GlcNAcylation upregulated the expression of multiple SM contractile markers in the aortic vasculature of both STZ-induced hyperglycemic smOGT^KO^ and Western diet-fed hyperglycemic smOGT^KO^ApoE^-/-^ mice. Our data lends strong support to the notion that OGT promotes VSMC de-differentiation in response to hyperglycemia. Characterization of lesion complexity further demonstrated an attenuation in SMC abundance and leukocyte infiltration in aortic root of hyperglycemic smOGT^KO^ApoE^-/-^ mice vs. hyperglycemic smOGT^WT^ApoE^-/-^ animals. This was accompanied by downregulation of the proliferation marker PCNA expression in hyperglycemic ApoE^-/-^ with SMC-targeted OGT deletion. Our results that smooth muscle OGT ablation attenuates cellular proliferation is in accord with earlier findings [35,53,54]. In the context of these reports, our results implicate a beneficial impact of SMC-targeted deficiency of OGT signaling in hyperglycemia-induced VSMC proatherogenic responses. Collectively, the current work underscores a direct regulatory role of SMC-derived OGT in lesion pathogenesis in diabetes.

Extracellular signal-regulated kinase 1/2 (ERK1/2), a member of the mitogen-activated protein kinase (MAPK) family, participates in many cellular programs including cell proliferation, differentiation, motility, and cell death [55]. There is accumulating evidence emphasizing a potential crosstalk between O-GlcNAcylation and ERK1/2 signaling pathways, with elevated O-GlcNAc levels stimulating ERK1/2 [56,57]. Earlier studies have also suggested that MAPK signaling regulates VSMC phenotypic transition, attributing to SMC heterogeneity and vascular pathology [58,59]. Consistent with these findings, we observed a significant reduction in pERK expression, a key signaling mediator of VSMC growth, in aortic lysates derived from smOGT^KO^ApoE^-/-^ vs. smOGT^WT^ApoE^-/-^ mice with intact OGT following Western diet-induced hyperglycemia. Concomitant to reduced ERK phosphorylation, SMC-targeted OGT deletion also attenuated proinflammatory markers (*Il1β* and *Il6*) known to be typically elevated in diabetic vascular pathology. Contrary to these results, Zhao et. al. previously revealed increased intestinal epithelial hypertrophy, epithelial hyperplasia, and mucosal thickness in intestinal epithelial-specific OGT knockout (Vil-OgtKO) mice, coupled with increased inflammatory gene expression (*Tnfα*, *Il6*, *Il1β*) [60]. The observed discrepancy in these findings may be due to cell-specific responses of OGT gene silencing explored in two diverse cell types (SMC vs. epithelial cells), attributing to disparate pathologies.

Accumulating literature suggests that VSMC phenotypic modulation is a function of numerous factors, including inflammatory mediators, cytokines and growth factors, and extracellular matrix proteins [61,62]. Increasing evidence further demonstrates that a triad of transcription factors (TFs) and co-regulators mediate the transcriptional regulation of de-differentiated VSMC gene expression, while disrupting the normal genetic program for VSMC contractile function. Interaction of one such TF called the serum response factor (SRF) with its co-factor myocardin (MYOCD) is recognized as the central regulator of VSMC contractile gene transcription [63]. In contrast, multiple transcriptional repressors including YY1, ELK-1 and KLF4 pathways block SRF-MYOCD interaction, repressing the SMC contractile phenotype [64,65,66]. Despite extensive efforts, significant gaps persist in our understanding of how diabetes modulates the molecular pathways mediating VSMC phenotypic transition. Growing data indicate that O-GlcNAcylation of transcriptional proteins controls numerous cell functions relevant to diverse pathological conditions. Our current findings prompt us to speculate that YY1- and SRF-dependent pathways play a regulatory role in OGT-mediated SMC de-differentiation induced by hyperglycemia, promoting accelerated atherosclerotic lesion formation in diabetes. Recruitment of SRF-MYOCD complex on the CArG box, a highly conserved cis-regulatory element (CC(A/T)6GG) found in the promoters of most SMC-specific genes, stimulates multiple SMC contractile gene transcription (*Sm22α*, *Acta2*, *Sm-mhc*) [64]. Previous literature indicates that the nuclear protein SRF can mediate disparate target gene expression, stemming from its ability to associate with alternating classes of co-factors [67,68]. Indeed, an earlier study identified SRF as a key regulator of VSMC proliferation and senescence [69]. Our data are comparable to these previous observations; specifically, in the current work, SMC-specific OGT deletion reduced SRF expression concomitant to attenuated PCNA expression in the aortic vasculature of smOGT^KO^ApoE^-/-^ following Western diet-induced hyperglycemia. While these data accentuate the contribution of SRF in VSMC proliferation, our findings suggest a novel role of OGT-mediated O-GlcNAcylation regulating this process. YY1, on the other hand, is a ubiquitous multifunctional TF belonging to the GLI- Krüppel family of zinc-finger proteins. YY1 was reported to bind to the CArG box in the SM22 promoter and operate as a transcriptional activator in SMCs [70]; on the other hand, YY1 has also been shown to repress smooth muscle promoter activity by competitively displacing MYOCD from SRF inhibiting the MYOCD/SRF/CArG box-mediated gene transcription [71,72]. A previous study demonstrated increased YY1 expression in vascular SMC in response to mechanical vascular damage, thereby blocking SMC replication in vitro [73] More recently, Zheng et al. (2020) reported that vascular injury induces YY1 expression in carotid artery medial SMCs in association with reduced transcription of SMC differentiation marker [64]. YY1 overexpression was also shown to decrease VSMC cell growth and migration [74]. Accordingly, the existing literature and our current findings support an inhibitory role of YY1 on SMC differentiation, possibly acting as a transcriptional repressor. As such, we speculate that interaction of YY1- and SRF-dependent pathways drive OGT-mediated SMC transformation to a de-differentiated phenotype in response to hyperglycemia. While our data suggests a putative crosstalk between YY1/SRF-related mechanisms and OGT-mediated VSMC de-differentiation, additional studies are warranted to elucidate the molecular pathways that link OGT to YY1 and SRF in regulation of VSMC phenotypic transformation in diabetes. We further posit that OGT-mediated O-GlcNAcylation of YY1 activates SRF-dependent pathways and binding partners advancing VSMC cell cycle progression, concomitant to O-GlcNAc modification of alternate transcriptional regulator(s) promoting VSMC-to-macrophage fate switch and inflammatory VSMC phenotype in diabetes.

One of the limitations of this study relates to the y-linked *Cre* murine driver employed, which restricted interrogation of sex-based differences in OGT-mediated SMC de-differentiation. Given that SMCs play a prominent role in vascular pathology, and vascular disease accounts for multiple cardiovascular anomalies striking men and women alike, it would be important to decode the molecular link between OGT and diabetic vasculopathy across both sexes. To this end, future investigation currently underway in our lab will utilize a VSMC-restricted Cre driver murine line (*Itga8-CreER^T2^*) to enable sex-specific studies and to selectively knockout OGT in VSMCs of vessel wall while sparing visceral SMC [75]. Notably, using VSMC-specific reporter mice combined with VSMC-specific OGT deletion, additional work will interrogate the spatiotemporal contribution of OGT-mediated O-GlcNAcylation in VSMC fate switch to diseased phenotypes and delineate the underlying molecular mechanisms. These studies are currently being pursued in our laboratory.

In conclusion, the present study uncovers a protective outcome of SMC-specific OGT deletion in hyperglycemia-induced atherosclerosis (Figure 10). Our data underscores a conceptual link between OGT-mediated O-GlcNAcylation and SMC de-differentiation and highlights smooth muscle OGT as a putative target of hyperglycemia-induced vasculopathy. Future work will determine whether OGT-mediated O-GlcNAcylation drives VSMC fate transition during lesion pathogenesis in diabetes. Additionally, elucidation of the temporal course and contribution of smooth muscle OGT signaling to lesion regression in diabetes is warranted.

## 4. Materials and Methods

### 4.1. Mouse Models

All animal procedures including mice euthanasia were conducted according to protocols reviewed and annually approved by the Institutional Animal Care and Use Committee at Northeast Ohio Medical University, in accordance with the NIH guidelines for the Care and Use of Laboratory Animals. SMC-specific *Ogt* knockout mice (smOGT^KO^) were generated by crossing female homozygous *Ogt*-floxed (OGT^fl/fl^) mice (B6.129-*Ogt^tm1Gwh^*/J; JAX # 004860) with male tamoxifen (Tmx)-inducible hemizygous *Myh11-CreER^T2^* mice (Cre^tg^; JAX # 019079) purchased from The Jackson Laboratories (Bar Harbor, ME). The OGT^fl/fl^ mice have loxP restriction sites on either side of the exon encoding amino acids 206–232 of the x-linked *Ogt* gene. The *Myh11-CreER^T2^* mice express CreER^T2^ under the control of the mouse smooth muscle myosin heavy polypeptide 11 (*Myh11*, aka *Sm-mhc*) promoter. The resulting OGT^fl/Y^/Cre^tg^ male mice (produced in F1 generation) were initially used for Cre recombinase activation to test for the successful generation of SMC-specific knockouts. Please note that, because the *Ogt* floxed gene is x-linked, all male mice produced were OGT^fl/Y^ and females were OGT^fl/fl^. Given that the *Cre* transgene is Y-linked, only male mice carried the *Cre* transgene. To generate the experimental mice for this study, male OGT^fl/Y^/Cre^tg^ mice were bred with female OGT^fl/+^; the resulting OGT^fl/Y^/Cre^tg^ and age-matched OGT^+/Y^/Cre^tg^ littermate mice were used for Cre recombinase activation. In a concurrent study, OGT^fl/Y^/Cre^tg^ male mice were cross-bred with ApoE^-/-^ female mice (JAX stock #002052) to generate Cre^tg^/OGT^fl/Y^/ApoE^-/-^ and age-matched Cre^tg^/OGT^+/Y^/ApoE^-/-^ littermates subsequently utilized for Cre recombination. OGT^F^, Cre^tg^ and ApoE^-/-^ genotypes were confirmed by polymerase chain reaction, zx per modification of established protocols (The Jackson Laboratory). Primer sequences, expected size of PCR products and PCR conditions used for mice genotyping are provided in Appendix A. All mice utilized in this study were on a C57BL/6J background, housed in a pathogen-free environment and maintained on a 12 h:12 h light/dark cycle.

### 4.2. Study Design

In the first study, mice weaned at 4-wks-age were maintained on standard laboratory diet (Purina LabDiet 5008) provided ad libitum until the study end point. Briefly, upon weaning, mice were randomly allocated to the following groups: (i) tamoxifen-treated Cre^tg^/OGT^fl/Y^ (smOGT^KO^), (ii) vehicle-treated Cre^tg^/OGT^fl/Y^ (smOGT^WT^), or (iii) tamoxifen-treated Cre^tg^/OGT^+/Y^ (smOGT^WT^). For Cre recombinase activation, 6-wks-old male mice were treated with tamoxifen (Tmx, 60 mg/kg/day) or vehicle (corn oil) intraperitoneally once daily for 5 consecutive days. After allowing for one-week ‘washout’, mice were injected with either low-dose STZ (50 mg/kg/day) or sodium citrate buffer, pH7.4 (vehicle control) intraperitoneally once daily for five consecutive days, beginning at 8-wks-age, for the induction of hyperglycemia. Ten days after the last STZ injection, blood samples collected by lateral tail incision were used for glucose estimation using a one-touch glucometer. Mice with non-fasted blood glucose levels ≥ 250 mg/dL were identified as hyperglycemic and assigned to the study for comparison with the normoglycemic genotypes. In a parallel study, 6-wks-old Cre^tg^/OGT^fl/Y^/ApoE^-/-^ and age-matched Cre^tg^/OGT^+/Y^/ApoE^-/-^ littermate male mice were subjected to Tmx-induced Cre recombination as described above, resulting in development of smOGT^KO^ and smOGT^WT^ mice on ApoE^-/-^ backgrounds, respectively. At 8-wks-age, mice were subjected to a Western Diet (TD88137, Teklad, Harlan) feeding regimen. In each study, body weight and non-fasted blood glucose levels were monitored every two weeks in all animals. After an overnight fast, mice were harvested at 14–16 weeks of age for blood and tissue collection following euthanasia using Fatal Plus.

### 4.3. Plasma Lipid Analyses

Plasma separated from blood samples was utilized for estimation of plasma total cholesterol and total triglyceride levels using Infinity Reagents (Thermo Fisher Scientific, Middletown, VA, USA).

### 4.4. Glucose Tolerance Test

Following an overnight fast, a lateral tail incision was made to obtain an initial blood sample to measure the baseline plasma glucose in each mouse. This was followed by a single intraperitoneal injection of sterile glucose solution (2 g/Kg body weight) to conduct a Glucose Tolerance Test (GTT). Blood samples were periodically collected 15, 30, 60, 90, and 120 min after the glucose injection. For each time point, a hand-held glucometer was utilized for the measurement of blood glucose levels.

### 4.5. Echocardiographic Analysis of Cardiac Function

At 16-wks-age prior to harvest, cardiac function was measured in each mouse via echocardiography using the Vevo 770 system (VisualSonics), equipped with a 710B-075 transducer (20–30 MHz) specifically designed for small animal investigations at a frame rate of 40–60 Hz, as described earlier [76]. Images were collected in M-mode from the parasternal short axis view at mid-papillary muscle level. Briefly, each mouse was placed on a controlled heating platform, made for small animal echocardiography, following anesthesia. A nose cone was utilized to deliver 1–2% isoflurane at 0.5 L/min with oxygen for induction of anesthesia. Following removal of chest hair, Aquasonic 100 gel (Parker Laboratories) pre-warmed to 37 °C was applied to the chest followed by assessment of cardiac function. All echocardiographic measurements and calculations were performed offline using the Vevo770/3.0.0 software. Five cycles were used to average all measurements.

### 4.6. Metabolic Phenotyping

Prior to harvest, a subset of mice was subjected to metabolic phenotyping using the Comprehensive Lab Animal Monitoring System (CLAMS). Briefly, mice were kept individually in sealed Plexiglass cages through which fresh room air was provided at 0.5 L/min for the entire duration of the experiment, which included 24 h of acclimatization followed by at least 24 h of normal recording. During this time, the animals had unlimited access to food and water and indirect calorimetry was used to assess their metabolic performance. For each mouse, oxygen consumption (VO_2_), carbon dioxide production (VCO_2_), respiratory exchange ratio (RER [VCO_2_/VO_2_]), and energy production were measured and recorded over the course of 24 h. Furthermore, an EchoMRI 3-in-1 body composition analyzer was used to measure both fat and lean tissue mass in live mice prior to CLAMS.

### 4.7. Aortic Root Morphometry

At the study endpoint, mouse hearts were removed after PBS perfusion. After a brief PBS wash, individual hearts were OCT-embedded and stored at −80 °C for further processing. Following dissection of the OCT-embedded hearts, serial sections (8 microns) of the aortic root were obtained, as reported earlier [77]. For all morphometric measurements, serial sections within 100–150 microns from the valve leaflet were utilized. Moreover, care was taken to ensure that only sections from comparable areas of the aortic root across all treatment groups were included for further staining and quantification. Concurrent aortic root sections derived from each mouse were stained with 0.5% Hematoxylin and Eosin (H & E) and Oil red O (ORO) solutions to detect plaque area and lipid burden, respectively. Furthermore, hematoxylin was used as a counterstain for all ORO-stained sections. Tissue sections were mounted using the DPX mounting media (Electron Microscopy Sciences) and images were captured at 10X magnification using the Olympus BX61VS microscope. For quantitative morphometry, we utilized a total of 7 mice for each genotype, and at least 30 sections from each group were analyzed.

### 4.8. Immunohistochemistry

Serial aortic root sections derived from each animal were subjected to immunohistochemistry using CD45 (Bioss, Woburn, MA, USA) and ACTA2 (Sigma, St. Louis, MO, USA) antibodies. Briefly, tissue sections were incubated in ice-cold acetone for 5–10 min followed by blocking with 5% donkey or goat serum at room temperature for an hour. After an overnight incubation at 4 °C with primary antibodies (1:200) diluted in the blocking solution, sections were incubated with Alexa Flour 594 donkey anti-rabbit IgG secondary antibody (1:400, Vectashield, Vector Laboratories) at room temperature for 1 h. To confirm non-specific staining, identical root sections were concurrently incubated either in the absence of the corresponding primary antibody or with species-specific isotype-matched IgG control antibody relevant to the primary antibody. Images were acquired using the Olympus fluorescence IX71 microscope at 10X magnification, with an identical set of parameters applied across all sections, specific for each antibody.

### 4.9. Immunoblotting and Quantitative Real Time PCR

Aortic vessels isolated from the mice were utilized in immunoblotting. Briefly, aortic protein lysates were prepared in RIPA lysis buffer (Boston BioProducts) and protein concentration was measured using the BCA Gold protein assay (Thermo Fisher Scientific, Middletown, VA, USA). Equal protein amounts (25 μg) of each sample were resolved on an 8% SDS-PAGE gel followed by a wet transfer to PVDF membranes. Immunoblotting was performed using the following antibodies: OGT, ACTA2, YY1, SRF, pERK and tERK (1:1000, Cell Signaling, Danvers, MA, USA); O-GlcNAc (RL2, 1:1000) and PCNA (1:500, Abcam, Waltham, MA, USA); LMOD1 and OGA (1:1000, Proteintech, Rosemont, IL, USA). Equal protein loading was confirmed by staining the membranes with Ponceau S, used as a loading control. All immunoblot images were captured using Amersham ImageQuant 800 and densitometric analyses were performed using Image J. For mRNA analysis, total RNA was extracted from the aortic vessels isolated from each mouse using Trizol Reagent (Sigma−Aldrich, St. Louis, MO, USA), as per manufacturer’s instructions. RNA concentrations were determined using a Nanodrop spectrophotometer (Thermo Fisher). Two micrograms of total mRNA were converted into cDNA using the RETROscript Reverse Transcription Kit (Thermo Fisher). Quantitative RT-PCR was performed using 2–5 μL of the reverse transcribed cDNA, 0.5 nM of specific forward and reverse primers and 1X PowerUp^TM^ SYBR Green or Taqman PCR master mix (Applied Biosystems) in a CFX96 PCR system (Bio-Rad Laboratories, Hercules, CA, USA). mRNA expression levels were normalized using the housekeeping genes 36b4 and 18S. All RT and PCR reactions were conducted in duplicate for each sample with and without RT as controls; cycle threshold (C_t_) values were converted to relative gene expression levels using the 2−^ΔΔC(t)^ method. Primer sequences and PCR cycling conditions used are provided in Appendix A.

### 4.10. Image Quantification

For each aortic root image, the lesion area was defined by the internal elastic lamina to the luminal edge of the lesion. Using the polygon selection tool in Image J, this region was cropped and saved as a new image file for subsequent analyses. For ORO-stained and immunofluorescent images, the total lesion area and stained area for each image was measured using the color thresholding option in Image J. For H&E-stained images, line tracings were drawn in each aortic root image to mark the area enclosed by the internal elastic lamina (total aortic root area) and the luminal edge of the lesions (aortic root luminal area). Plaque area was then determined by subtracting the aortic root luminal area from the total aortic root area. For immunohistochemistry, specific positive staining was expressed as a percentage of the total lesion area. All image quantifications were performed by team members blinded to the identity of the sections. For lipid burden, ORO-positive area was measured in μm^2^; for H&E images, plaque area was measured in mm^2^; for CD45 and ACTA2 images, area was measured in sq. pixels.

### 4.11. Statistical Analysis

Statistical analyses were performed using GraphPad Prism version 8. Each data set was assessed for normality (Shapiro-Wilks; Kolmogorov-Smirnov) and homoscedasticity (Brown-Forsythe) prior to further analysis. When the assumption of normality was met, parametric testing methods were used, whereas non-parametric methods were employed when the conditions of normality were violated. Ordinary one-way ANOVA was used for normally distributed data with equal variance followed by Tukey HSD post-hoc test. For normally distributed data with unequal variance, Welch ANOVA was applied followed by Games-Howell or Dunnett’s T3 post-hoc test. Mann-Whitney and Kruskal-Wallis were used as the non-parametric counterparts of Student’s t-test and one-way ANOVA, respectively; Kruskal-Wallis was followed by the Dunn’s post-hoc test. Statistical significance was considered at *p* ≤ 0.05. Each experiment was repeated at least three times in an independent setting, with two to five replicates for each treatment. In immunofluorescence experiments, six to eight independent field images were acquired for each individual treatment in a single experiment. For aortic root morphometry and immunohistochemistry, four to five sections per mouse and three to eight mice per group were examined. Image quantifications were performed as described above. For all immunoblots, lane images depict proteins loaded and detected on a single blot. However, for specific immunoblots as indicated in the corresponding figure legends, lanes were rearranged for the clarity of presentation. All data are presented as fold-increase vs. controls. Values are expressed as Mean ± Standard error of Mean (SEM).

## Figures and Tables

**Figure 1 ijms-24-07899-f001:**
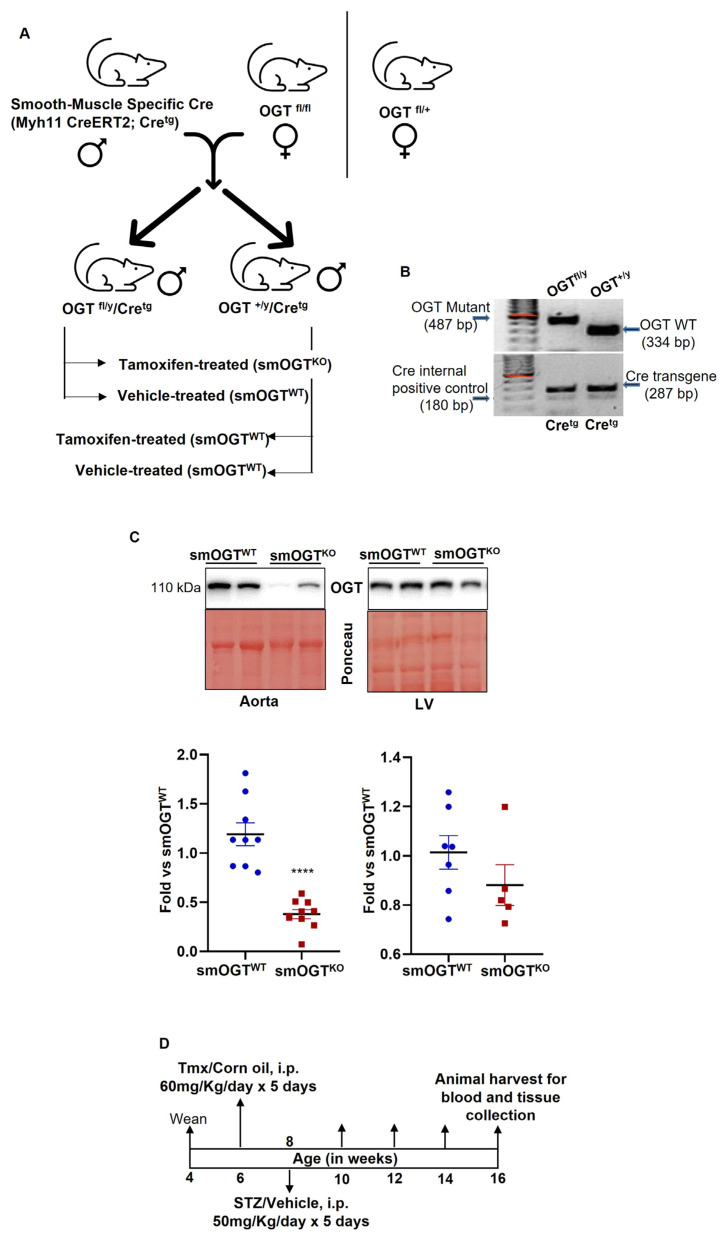
Validation of inducible SMC-specific OGT knockout mice. (**A**) Breeding strategy for generation of smOGT^KO^ mice, (**B**) PCR genotyping data of genomics DNA, (**C**) representative immunoblots and summary graphs showing OGT expression in aorta vs. left ventricle (LV) isolated from smOGT^KO^ vs. smOGT^WT^ mice. Values represent fold-change vs. smOGT^WT^ normalized to Ponceau S (for total protein staining); error bars denote SEM values. *n* = 5–9 per group, **** *p* < 0.0001, (**D**) study timeline for STZ-treated hyperglycemic smOGT^WT^ vs. smOGT^KO^ mice.

**Figure 2 ijms-24-07899-f002:**
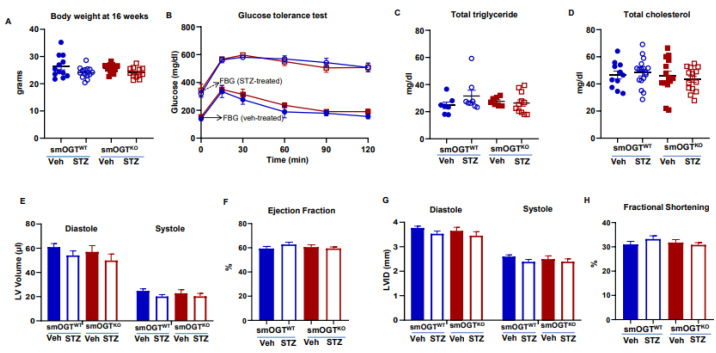
Metabolic and cardiac phenotypes of SMC-specific OGT knockout mice under basal and STZ-induced hyperglycemic conditions. Male smOGT^WT^ and smOGT^KO^ mice were treated with STZ or sodium citrate (vehicle control), as shown in the study timeline in figure legend 1. Shown are (**A**) body weight, (**B**) Glucose tolerance test (GTT), (**C**) Total triglyceride, (**D**) Total cholesterol, (**E**) LV volume, (**F**) ejection fraction, (**G**) LV internal diameter (LVID) and (**H**) fractional shortening of vehicle- and STZ-treated smOGT^WT^ and smOGT^KO^ mice at 16-wks-age. For (**B**), solid and dashed arrows denote Fasting Blood Glucose (FBG) values for Vehicle- and STZ-treated mice, respectively. For (**A**–**D**), error bars denote SEM values; for (**E**–**G**), results are presented as mean ± SEM, *n* > 5 per treatment group.

**Figure 3 ijms-24-07899-f003:**
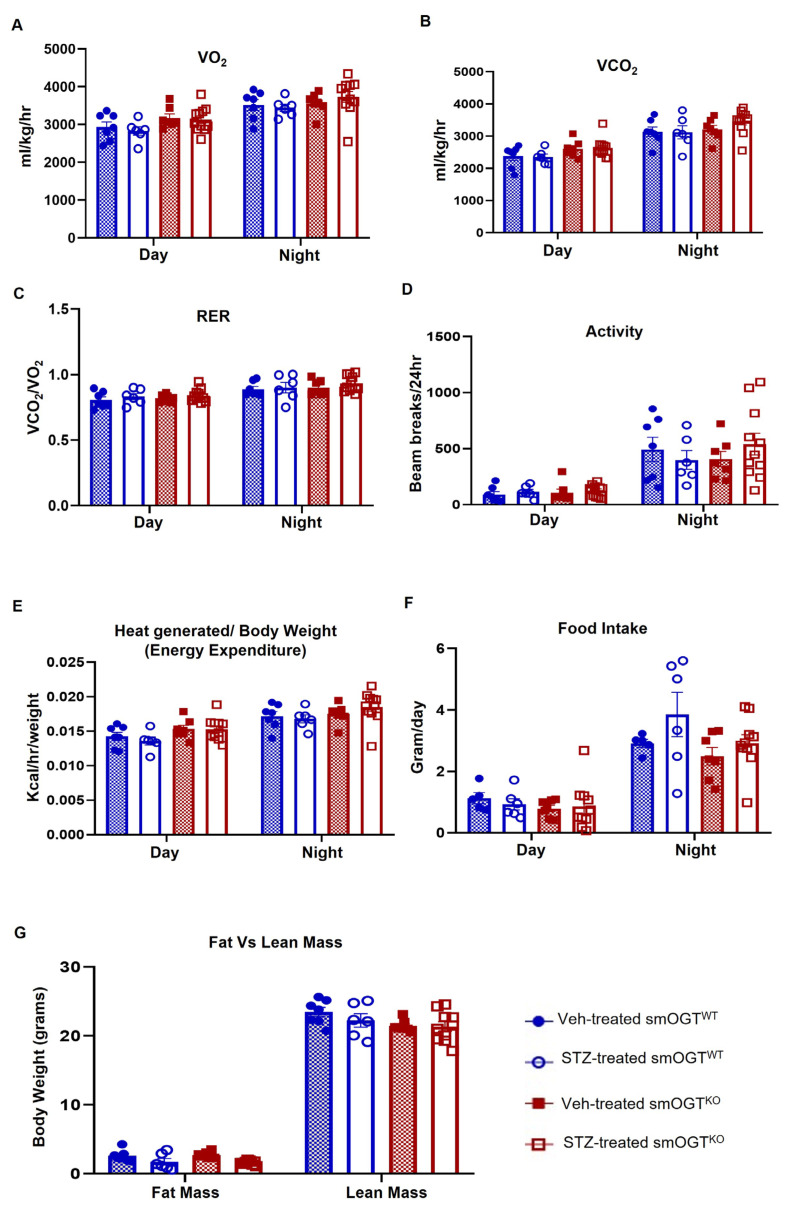
Indirect calorimetry of SMC-specific OGT knockout mice under basal and STZ-induced hyperglycemia via CLAMS. Male smOGT^KO^ and smOGT^WT^ mice were treated with or without STZ and subjected to metabolic phenotyping using CLAMS, as described in Methods. (**A**) Oxygen consumption (VO_2_), (**B**) carbon dioxide production (VCO_2_), (**C**) respiratory exchange ratio (RER), (**D**) physical activity, (**E**) energy expenditure, (**F**) food intake and (**G**) fat and lean mass. Values represent mean ± SEM, *n* = at least 5 animals per group.

**Figure 4 ijms-24-07899-f004:**
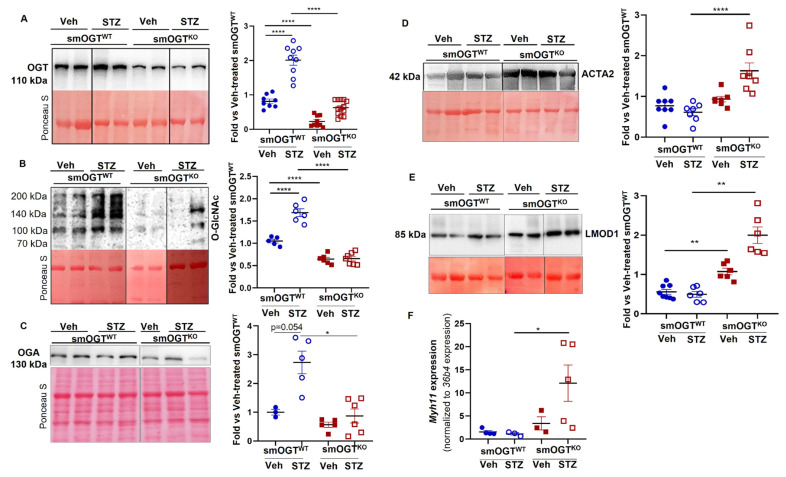
SMC-specific OGT deletion increases SM contractile marker expression in aortic vessels of STZ-induced hyperglycemic mice in vivo. (**A**–**E**) Immunoblotting of aortic lysates derived from vehicle- and STZ-treated smOGT^WT^ and smOGT^KO^ mice. Shown are representative immunoblots (left panels) and corresponding summary graphs (right panels) for (**A**) OGT, (**B**) O-GlcNAc, (**C**) OGA, (**D**) ACTA2 and (**E**) LMOD1 expression. Values represent fold-change vs. vehicle-treated smOGT^WT^ normalized to Ponceau S (for total protein staining); *n* = 3–9 mice per group, * *p* < 0.05, ** *p* < 0.005, **** *p* < 0.0001. Lane images show proteins identified and detected on a single immunoblot (see Appendix A for full blots); however, lanes were rearranged for the purpose of clarity of presentation. (**F**) Relative *Myh11* mRNA expression. *n* = 3–5 mice per group, * *p* < 0.05. Error bars denote SEM values.

**Figure 5 ijms-24-07899-f005:**
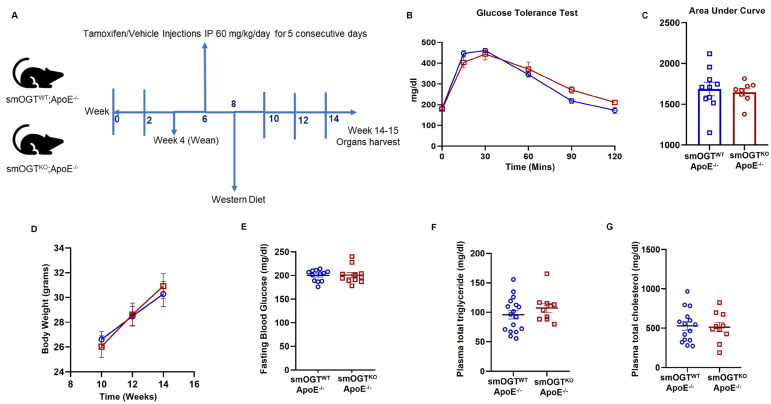
Metabolic profile of Western diet−fed ApoE^-/-^ mice with and without SMC-specific OGT deletion. Male smOGT^WT^;ApoE^-/-^ and smOGT^KO^;ApoE^-/-^ mice were subjected to Western diet feeding regimen for 6–7 wks. (**A**) Study timeline, (**B**) glucose tolerance test (GTT), (**C**) GTT Area under Curve, (**D**) body weight, (**E**) fasting blood glucose, (**F**) total triglyceride and (**G**) total cholesterol. Values represent mean ± SEM, *n* ≥ 10 mice per group.

**Figure 6 ijms-24-07899-f006:**
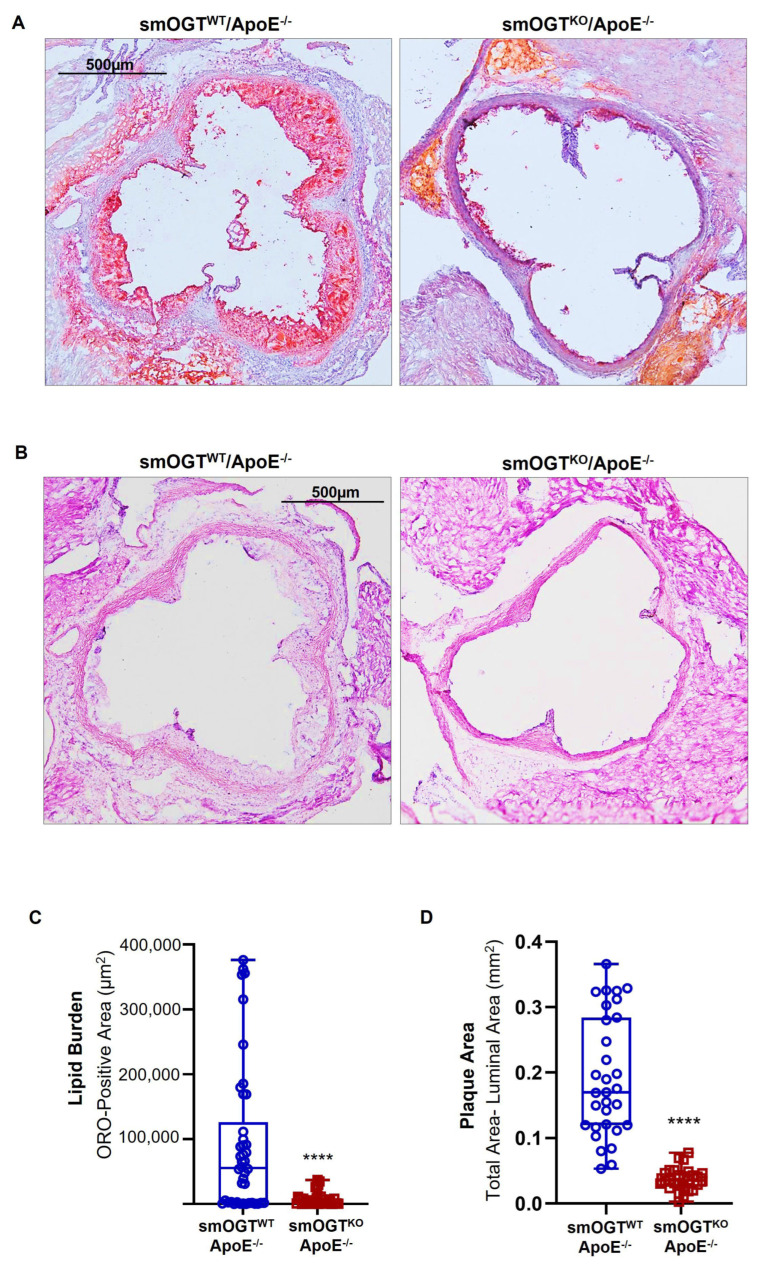
SMC-specific OGT deletion impedes lipid burden and plaque area in Western diet−fed ApoE^-/-^ mice. Shown are representative (**A**) Oil red O (ORO)- and (**B**) hematoxylin & eosin (H&E)-stained images of aortic root sections derived from 14-wks old Western diet-fed smOGT^WT^ApoE^-/-^ and smOGT^KO^ApoE^-/-^ mice. Corresponding summary graphs for lipid burden and plaque area are shown in (**C**,**D**). Results are illustrated in box and whisker plot format, with median values shown via the horizontal line across the center of the box; the ‘top’ and ‘bottom’ whiskers represent the ‘minimum to lower quartile value’ and ‘upper quartile to maximum data value’, respectively. Each value denotes lipid burden (**A**) and plaque area (**B**) of aortic root per section in each study group, *n* = 7 mice with at least 30 aortic root sections from each group. **** *p* < 0.0001.

**Figure 7 ijms-24-07899-f007:**
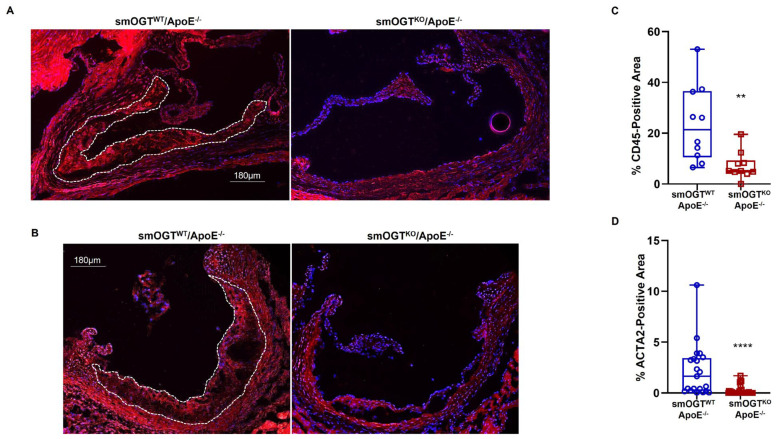
SMC-specific OGT deletion reduces inflammatory and smooth muscle cell content in Western diet-fed ApoE^-/-^ mice. Shown are representative immunofluorescent images depicting (**A**) CD45 and (**B**) ACTA2 expression in aortic root sections derived from Western diet-fed smOGT^WT^ApoE^-/-^ and smOGT^KO^ApoE^-/-^ mice. Corresponding summary graphs are shown in (**C**,**D**). White dotted line marks the lesion area used for quantification. Shown are %CD45 or %ACTA2-positive area of total lesion per section in each study group. Results are shown in box and whisker plot format, with median values denoted via the horizontal line across the center of the box; the ‘top’ and ‘bottom’ whiskers represent the ‘minimum to lower quartile value’ and ‘upper quartile to maximum data value’, respectively. *n* = 5–6 mice with 2–4 sections/aortic root/group. ** *p* < 0.005; **** *p* < 0.0001.

**Figure 8 ijms-24-07899-f008:**
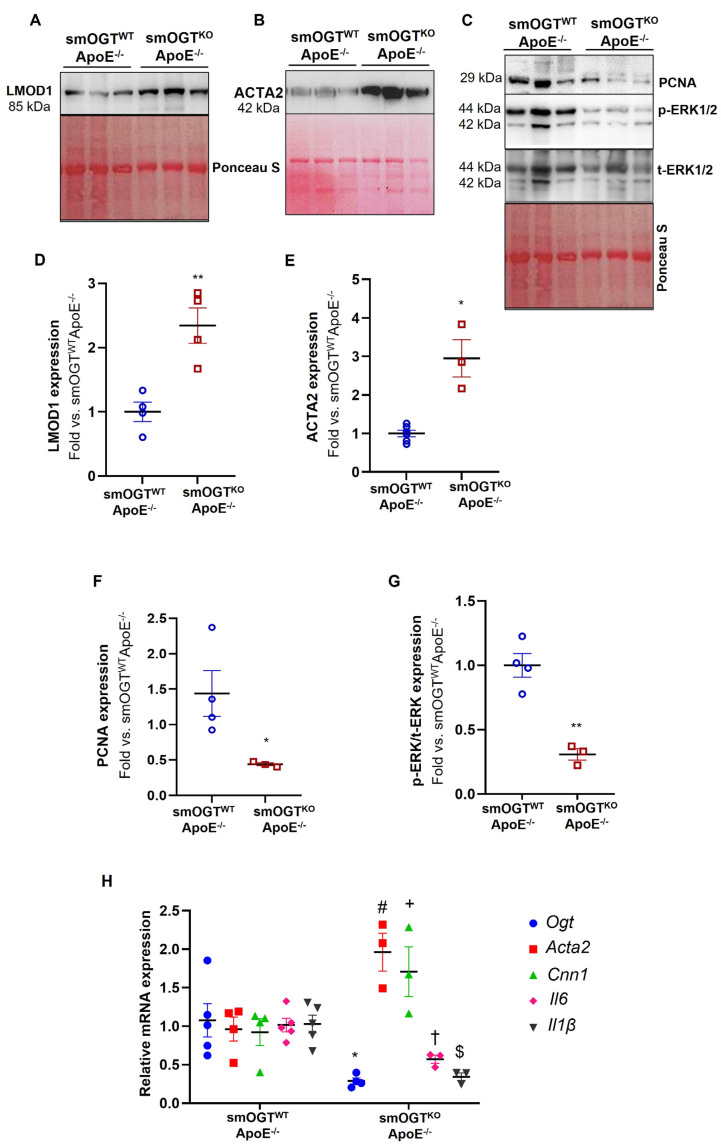
SMC-specific OGT deletion increases SM contractile marker expression concomitant to reduced proliferative and inflammatory marker expression in aortic vessels of Western diet−fed ApoE^-/-^ mice. Aortic lysates derived from Western diet-fed smOGT^WT^ApoE^-/-^ and smOGT^KO^ApoE^-/-^ mice were subjected to immunoblotting. Shown are (**A**–**C**) representative immunoblots and (**D**–**G**) corresponding summary graphs depicting LMOD1, ACTA2, PCNA and pERK/tERK expression. Values represent fold-change vs. smOGT^WT^ApoE^-/-^ normalized to Ponceau S (for total protein staining). (**H**) Relative mRNA expression of *Ogt*, *Acta2*, *Cnn1*, *Il6* and *Il1β*; *n* = 3–5 mice per group. Error bars denote SEM values; * *p* < 0.05, ** *p* < 0.005, ^#^
*p* < 0.05, ^+^
*p* < 0.05, ^†^
*p* < 0.05, ^$^
*p* < 0.05.

**Figure 9 ijms-24-07899-f009:**
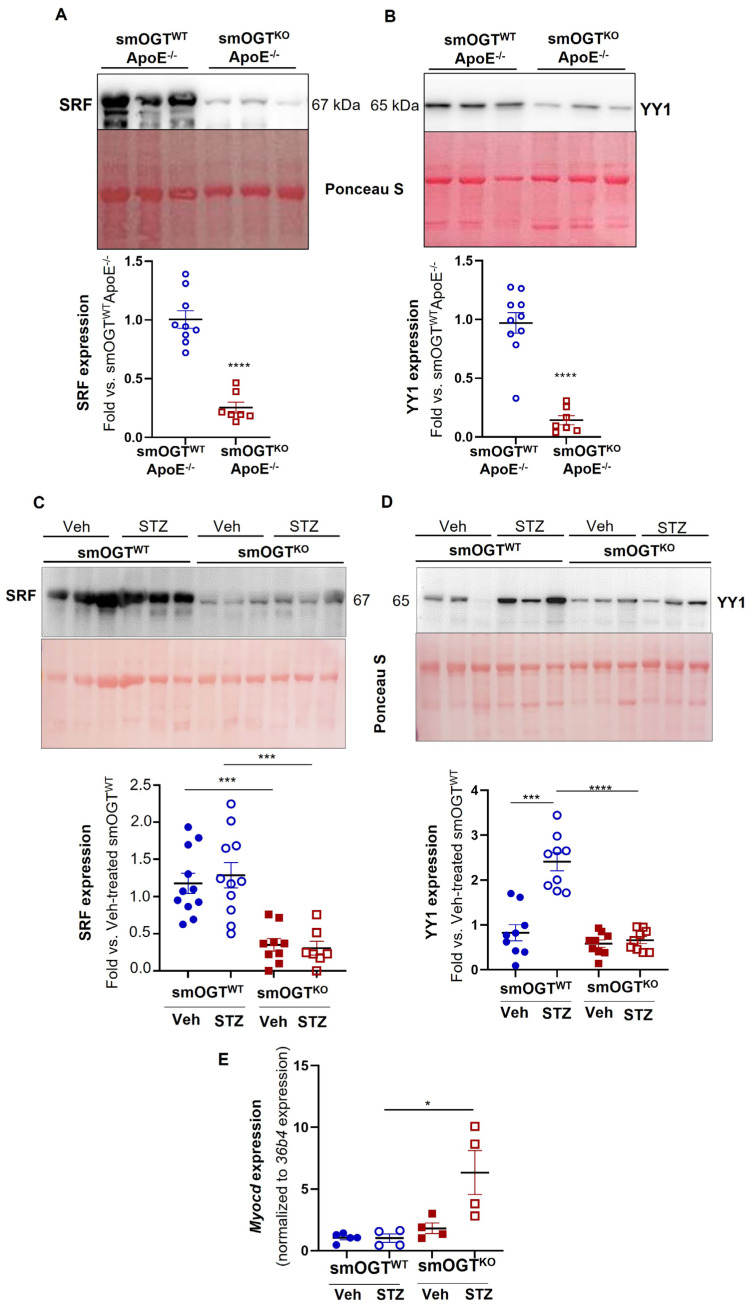
SMC-specific OGT deletion attenuates YY1 and SRF expression in aortic vasculature of Western diet−fed ApoE^-/-^ and STZ-induced hyperglycemic mice in vivo. Shown are immunoblotting of aortic lysates from (**A**,**B**) Western diet-fed smOGT^WT^ApoE^-/-^ and smOGT^KO^ApoE^-/-^ and (**C**,**D**) STZ-treated hyperglycemic and non-STZ treated non-hyperglycemic smOGT^WT^ and smOGT^KO^ mice. The representative immunoblot images and corresponding summary graphs for (**A**,**C**) SRF and (**B**,**D**) YY1 expression are shown. Values denote (**A**,**B**) fold-change vs. smOGT^WT^ApoE^-/-^ and (**C**,**D**) fold-change vs. vehicle-treated smOGT^WT^ normalized to Ponceau (for total protein staining), *n* = 7–11 mice per group; *** *p* < 0.0005, **** *p* < 0.0001. (**E**) Shown are the relative *Myocd* mRNA expression in aortic lysates derived from STZ-induced hyperglycemic and vehicle-treated non-hyperglycemic smOGT^WT^ and smOGT^KO^ mice. *n* = 4–5 mice per group, * *p* < 0.05. Error bars denote SEM values.

**Figure 10 ijms-24-07899-f010:**
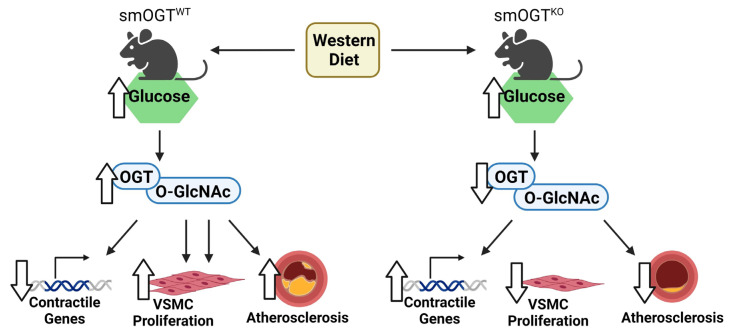
Summary of key findings. In Western diet-fed hyperglycemic smOGT^WT^ApoE^-/-^ mice (**left** panel), increased OGT-mediated O-GlcNAcylation promotes atherosclerotic lesion formation accompanied with increased VSMC proliferation and reduced SM contractile gene expression. On the other hand, in Western diet-fed hyperglycemic smOGT^KO^ApoE^-/-^ mice (**right** panel), reduced OGT-mediated O-GlcNAcylation impedes atherosclerotic lesion formation accompanied with reduced VSMC proliferation and augmented SM contractile gene expression. Created with BioRender.com (accessed on 14 April 2023).

## Data Availability

The data that support the findings of this study are available from the corresponding author upon reasonable request.

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
