# Peer review of "Deletion of Smooth Muscle O-GlcNAc Transferase Prevents Development of Atherosclerosis in Western Diet-Fed Hyperglycemic ApoE-/- Mice In Vivo"

_ijms, 2023, doi:10.3390/ijms24097899_

Round 1

Reviewer 1 Report

The manuscript by Khanal…Raman et al describes a full biological study wherein the key regulatory protein O-GlcNAc transferase is deleted in smooth muscle and tested in an ApoE-null mouse for atherosclerosis phenotype effects.

This study builds from earlier work by the authors wherein their mouse model was used and characterized for selective knockout of smooth muscle OGT vs. cardiac muscle OGT, which is an important advantage of the author’s system vs. previously developed OGT knockout muscle models. In the current study, the ApoE-null mice were subjected to high fat diet for 6-8 weeks before smooth muscle proliferation, contractile, and lipid markers were stained. The goal was to implicate the effect of OGT (via its knockout) on atherosclerotic lesion formation in response to hyperglycemia via the high-fat diet and hyperlipidemia via the ApoE-null setting.

The data presents a well-characterized system detailing the effects of these two parameters on smooth muscle functions. A few additional experiments and comments follow to be ready for publication:

Fig 1 or 2 -> what is the fasting or random blood glucose level of the model? Please add (it is stated in the text but now shown)

Before Figure 4, explain in the text what ACTA2 and LMOD2 are doing and what the expected result of low/high levels are. I am not clear as to their significance, being somewhat outside of this field.

Figure 4 -> what does the OGA expression look like? It is interesting that OGT is reduced to <15-30% expression, but overall O-GlcNAc levels remain at ca. 50%. Does OGA level go down? (we use the Abcam antibody to study OGA, which works well for us, by the way)

Figure 4, 6, 7, 8, and 9 -> what do the error bars mean? SEM?

Figure 6 -> is there a typo in the y-axis of 6D?

All figures are at very low resolution. This plus super small text size makes them difficult to read. Please fix.

The methods section is clear and concise.

Overall the Intro and Results sections are clearly written.

The Discussion, on the other hand, becomes extremely difficult to follow. Quite a bit of literature is brought in, approximately 50-60 new references. The resulting summary is better suited for a review article beyond the scope of this paper, one that would include the current results if they are accepted. I suggest cleaning up the Discussion section to describe the data that was collected and its implications rather than placing it in the context of a much larger “review type” summary. I also very much suggest making a model figure to summarize the new knowledge the article gains.

With the few additional corrections (and OGA staining), and especially cleaning up the Discussion section, I recommend this paper to be accepted by IJMS after these revisions. The study is quite complete and shows an interesting role for hyperglycemia-driven signaling via O-GlcNAc and OGT.

Reviewer 2 Report

The manuscript by Saugat Khanal et al demonstrate the role of smooth muscle O-GlcNAc transferase on development of atherosclerosis through western diet-fed animal model.

1. In Fig 4A and B, OGT and O-GlcNAc protein expression were upregulated in the aortic vessels of smOGTWT mice following STZ-induction. But the residual OGT and O-GlcNAc protein expression in the KO mice has no significant change between Veh group and STZ group. Do you have any explanation?

2. Do the SMC-specific OGT mouses have any phenotype on lesion burden, inflammation and SM contractile marker expression in normal chow-fed hyperglycemic ApoE-/- model?

3. The scale bar are needed in Fig 7A and B.

4. In Fig 8C, total ERK is needed.

Minor editing of English language required.

Round 2

Reviewer 2 Report

The revised version has been largely improved. 

Minor editing of English language required